# Joint Optimization of Multi-User Partial Offloading Strategy and Resource Allocation Strategy in D2D-Enabled MEC

**DOI:** 10.3390/s23052565

**Published:** 2023-02-25

**Authors:** Dongping Yong, Ran Liu, Xiaolin Jia, Yajun Gu

**Affiliations:** 1Mobile Internet of Things and Radio Frequency Identification Technology Key Laboratory of Mianyang (MIOT&RFID), Mianyang 621010, China; 2RFID & IOT Laboratory, School of Computer Science and Technology, Southwest University of Science and Technology, Mianyang 621010, China; 3School of Information Engineering, Southwest University of Science and Technology, Mianyang 621010, China

**Keywords:** D2D communications, helpers selection, power allocation, offloading strategy

## Abstract

With the emergence of more and more computing-intensive and latency-sensitive applications, insufficient computing power and energy of user devices has become a common phenomenon. Mobile edge computing (MEC) is an effective solution to this phenomenon. MEC improves task execution efficiency by offloading some tasks to edge servers for execution. In this paper, we consider a device-to-device technology (D2D)-enabled MEC network communication model, and study the subtask offloading strategy and the transmitting power allocation strategy of users. The objective function is to minimize the weighted sum of the average completion delay and average energy consumption of users, which is a mixed integer nonlinear problem. We first propose an enhanced particle swarm optimization algorithm (EPSO) to optimize the transmit power allocation strategy. Then, we utilize the Genetic Algorithm (GA) to optimize the subtask offloading strategy. Finally, we propose an alternate optimization algorithm (EPSO-GA) to jointly optimize the transmit power allocation strategy and the subtask offloading strategy. The simulation results show that the EPSO-GA outperforms other comparative algorithms in terms of the average completion delay, average energy consumption, and average cost. In addition, no matter how the weight coefficients of delay and energy consumption change, the average cost of the EPSO-GA is the least.

## 1. Introduction

With the continuous development of the Internet of Things (IoT) and the fifth-generation mobile communication technology (5G), the number of mobile devices will show a gradual upward trend. Statista forecasts that the number of mobile devices will increase from 15.96 billion to 17.72 billion in 2023. In addition, the development of artificial intelligence, virtual reality, augmented reality, and autonomous driving also puts forward higher requirements for communication and computing resources. They need to process large amounts of data in a relatively short period of time. When processing data requires a long time and exceeds a certain tolerance range, the quality of user service experience will be reduced. However, due to the limitation of the size of the mobile device itself, the resources of the mobile devices are limited and it is difficult to meet the needs of users, thus computing offloading becomes an effective way. The traditional method is to upload the tasks to the cloud for computing. However, the cloud is far away from the user, so the communication delay of offloading tasks to the cloud server will be relatively longer. The emergence of MEC effectively solves this dilemma. MEC achieves the purpose of reducing latency and energy consumption by offloading tasks to edge servers that are closer to users. In [1,2,3,4,5], they are based on the single-user MEC scenario to jointly optimize the offloading strategy and the resource allocation strategy to minimize the delay or energy consumption.

However, with the development of IoT and 5G, the number of users is increasing, there are often multiple users with computationally intensive tasks in the MEC scene, and they share the resources. As a result, the formulation of the user offloading strategy and resource allocation strategy are challenging tasks. In [6,7,8,9,10,11,12], they are all based on the multi-user MEC scenario to jointly optimize the offloading strategy of the tasks and the resource allocation strategy of the devices to minimize the delay or energy consumption.

In the above research, the users only consider two offloading methods, either locally computing or offloading tasks to edge servers through wireless channels for execution. However, neither considers neighboring devices that are closer to the user. That is, users can utilize device-to-device technology (D2D) to offload tasks to neighboring devices with idle resources for computing. D2D technology supports users to offload some tasks to adjacent neighbor devices for execution. This offloading method reduces the execution delay and energy consumption of the tasks, and heightens the execution efficiency of the tasks. In addition, this offloading method improves the utilization of resources in the environment. However, the formulation of the task offloading strategy in D2D-enabled MEC communication networks is an extremely challenging problem. In [13,14,15,16,17,18], the authors optimize the task offloading strategy or the resource allocation strategy. In [13], the authors proposed an offloading framework based on D2D collaboration, and optimize the task offloading strategy based on the Lyapunov algorithm to minimize energy consumption. In [14], the Joint Computing Offload and Resource Allocation Optimization (JCORAO) scheme is proposed, which leads to the optimal offloading strategy and optimal resource allocation scheduling. In [15], the authors proposed an algorithm called mobility-aware task scheduling (MATS), which jointly optimized the task offloading strategy and power allocation strategy and realized the purpose of minimizing the delay. In [16], the authors proposed a heuristic algorithm based on a greedy algorithm, which jointly optimized the task allocation, communication rate, and computing frequency allocation and achieved the purpose of minimizing delay. In [17], the authors proposed an alternate optimization algorithm based on the knapsack problem and convex optimization techniques, which jointly optimize the subtask partial offload decision and transmission power allocation problem, thus reducing the weighted sum of delay and energy consumption. In [18], the authors jointly optimized the helpers selection, communication allocation, and computing resource allocation using convex optimization techniques and block coordinate descent, thus achieving a fast convergence of results and minimizing energy consumption. In [19], the authors proposed an improved whale optimization algorithm that achieves reductions in the objective function value, power consumption, and average resource allocation imbalance rate. In [20], the authors proposed a deep-reinforcement-learning-based offloading scheduler (DRL-OS), achieving lower average latency.

Although previous work has studied D2D-enabled MEC systems, most of the above studies are single user or do not consider the case that the user has multiple subtasks. In our work, we consider the formulation of the transmit power allocation strategy and subtask offloading strategy for multiple users with multiple subtasks under the D2D-enabled MEC communication model. It is challenging to obtain the optimal transmit power allocation strategy and the optimal subtask offloading strategy. Therefore, it is very necessary to study the transmit power allocation strategy and subtask offloading strategy under the D2D-enabled MEC communication model.

The main contributions of this paper are as follows:We jointly optimize the transmit power allocation strategy and the subtask offloading strategy to minimize the average cost under the D2D-enabled MEC communication system; the average cost refers to the weighted sum of the total completion delay and the average energy consumption of the users to complete all subtasks.We propose the enhanced particle swarm optimization algorithm (EPSO), which can optimize the transmit power allocation strategy. The simulation results show that epso can cost less than other algorithms.We propose an alternate optimization algorithm, EPSO-GA, that combines the EPSO algorithm and Genetic Algorithm (GA). The algorithm can jointly optimize the transmit power allocation strategy and subtask offloading strategy. The simulation results show that the algorithm can effectively reduce the average cost of the users.

The remainder of this paper is organized as follows. Section 2 presents the system model and problem formulation. Section 3 describes the joint optimization of the transmit power allocation strategy and the subtask offloading strategy. Section 4 presents the simulation results and performance analysis. Section 5 summarizes the work of this paper.

## 2. System Model and Problem Formulation

### 2.1. System Model

In this paper, we consider a D2D-enabled MEC system shown in Figure 1, which contains *M* users with *N* computationally intensive subtasks, one base stations equipped with an edge server, and *H* neighbor helpers. The sets of users, the sets of subtasks, and the sets of neighbor helpers are, respectively, expressed as: $M=\left\{1,2,\ldots ,m,\ldots ,M\right\}$, $N=\left\{1,2,\ldots ,n,\ldots ,N\right\}$, $H=\left\{1,2,\ldots h,\ldots ,H\right\}$. Among them, for idle devices and resource-rich devices in the scene, such as laptops, tablet computers, desktop computers and other devices can connect with users through device-to-device technology (D2D) to act as neighbor helpers to facilitate tasks execution. Let $f_{m}$, $f_{server}$ and $f_{h}$ denote the computing power of the *m*th user, the edge server, and the *h*th neighbor helper, respectively. The transmit power of each user is expressed as $P=\{P_{1},P_{2},\ldots ,P_{m},\ldots P_{M}\}$, $P_{m}$ is the data transmit power of the *m*th user. $D_{m,n}$ is an attribute of computationally intensive subtasks, representing the data size of the *n*th subtask in the *m*th user.

In this paper, the subtasks are independent of each other, so the offload decisions between the subtasks do not affect each other. Let $k_{m,n}$ be the offloading decision of the *n*th subtask in the *m*th user, then there are three decision schemes for $k_{m,n}$: (1) Local computing ($k_{m,n}=0$) indicates that the subtask executes by itself. (2) Edge computing ($k_{m,n}=1$) indicates that the subtask is offloaded to the edge server for computing over the cellular network. (3) D2D offloading ($k_{m,n}=2$) means that the subtask is offloaded to the neighbor helper that is closest to the user for execution. Since the execution result of the subtask is relatively small compared to the uploaded data, this paper does not consider the return delay of the subtasks.

In this paper, there are many neighbor helpers that each user can choose. Therefore, when the subtask needs to be offloaded to the neighbor helper, the above-mentioned offloading decision expression is not enough to accurately express the destination of offloading. Therefore, we redefine the offloading decision and express the offloading decision in binary form as follows:(1)$$X_{m,n,k}=\left\{0,1\right\}.$$

The above formula represents whether the *n*th subtask in the *m*th user is offloaded to the *k*th neighbor helper, $k\in \left\{0,1,2,\ldots K+1\right\}$. Among them, $X_{m,n,0}=1$ means that the subtask will be executed locally, while $X_{m,n,K+1}=1$ means that the subtask will be offloaded to the edge server for execution. When $X_{m,n,k}=1$, $k\in \left\{1,2,\ldots ,K\right\}$ means that the subtask will be offloaded to the *k*th neighbor helper for execution. Note that each subtask can only be executed on one device and cannot be split, as shown in Formula (Equation 2).
(2)$$\sum_{k=0}^{K+1}X_{m,n,k}=1.$$

In this paper, we will apply the following assumptions:The network is quasi-static, and the network status and user information can be obtained by referencing plugins;All devices are single-core, and a device can only execute one task at a time;When multiple users offload tasks to a device at the same time, the tasks are executed on a first-come-first-served basis;Each subtask is of the same type; the number of CPU cycles required to process 1 bit of data is 1500 cycles [21].The channel bandwidths are *B*.

### 2.2. Problem Formulation

#### 2.2.1. Local Execution

The total time spent on subtasks executed locally on the *m*th user is expressed as:(3)$$T_{m}^{l}=\sum_{n=1}^{N}\frac{X_{m,n,0}\times D_{m,n}\times c}{f_{m}}.$$
where $D_{m,n}$ is the data size of the *n*th subtask in the *m*th user, *c* is the number of cycles required to process 1 bit of data, and $f_{m}$ is the CPU speed of the *m*th user. The energy consumption of the *m*th user is expressed as:(4)$$E_{m}=k_{m}\times {\left(f_{m}\right)}^{3}\times T_{m}^{l}.$$
where the value of $k_{m}$ depends on the chip structure of the device and is a constant. We set it to $10^{-27}$ according to [22].

#### 2.2.2. Edge Execution

When the task is to be offloaded to the edge server through the cellular network, the data transfer rate from the *m*th user to the edge server is expressed as:(5)$$R_{m}^{MEC}=B\times log_{2}\left(1+\frac{P_{m}\times g_{m}^{l,MEC}}{\sigma^{2}}\right).$$
where *B* is the channel bandwidth between the *m*th user and the edge server, $g_{m}^{l,MEC}$ is the path loss from the *m*th user to the edge server, $\sigma^{2}$ is the Gaussian white noise power.
(6)$$g_{m}^{l,MEC}=\delta \times {\left(d_{m}^{l,MEC}\right)}^{-2}=\frac{\delta }{\left({\left(X_{m}-X_{server}\right)}^{2}+{\left(Y_{m}-Y_{server}\right)}^{2}\right)}.$$
where δ is the channel gain parameters at the distance 1 m, and $d_{m}^{l,MEC}$ is the distance from the *m*th user to the edge server. The distance between the user and edge server can be obtained using the Euclidean distance. $X_{m},Y_{m}$ are the abscissa and ordinate of the *m*th user, respectively. $X_{server},Y_{server}$ are the abscissa and ordinate of the edge server, respectively. The transmission delay of the subtasks from the *m*th user to the edge server is expressed as:(7)$$T_{m}^{t,MEC}=\frac{\sum_{n=1}^{N}X_{\left(m,n,K+1\right)}D_{m,n}}{R_{m}^{MEC}}.$$

The execution delay of the subtasks offloaded from the *m*th user to the edge server is expressed as:(8)$$T_{m}^{e,MEC}=\frac{\sum_{n=1}^{N}X_{\left(m,n,K+1\right)}D_{m,n}\times c}{f_{server}}.$$
where $f_{server}$ is the CPU speed of the edge server. The total delay consumed by the subtasks offloaded to the edge server in the *m*th user is expressed as:(9)$$T_{m}^{MEC}=T_{m}^{t,MEC}+T_{m}^{e,MEC}.$$

In the process of subtasks offloading, we only consider the energy consumption of the users carrying computationally intensive tasks. Therefore, the energy consumption of the subtasks executed at the edge server in the *m*th user is expressed as:(10)$$E_{m}^{MEC}=E_{m}^{t,MEC}=P_{m}\times T_{m}^{t,MEC}.$$
where $P_{m}$ is the transmit power of the *m*th user.

#### 2.2.3. Neighbor Helper Execution

The distance from the neighbor helpers to the users is less than the distance from the edge server to the users. The data transmission rate between the *m*th user and the *k*th neighbor helper is expressed as:(11)$$R_{m,k}^{D2D}=B\times log_{2}\left(1+\frac{P_{m}\times g_{m,k}^{D2D}}{\sigma^{2}}\right).$$
where $g_{m,k}^{D2D}$ is the path loss from the *m*th user to the *k*th neighbor helper, and its expression is as follows.
(12)$$g_{m,k}^{D2D}=\delta \times {\left(d_{m,k}^{D2D}\right)}^{-2}=\frac{\delta }{{\left(X_{m}-X_{k}^{D2D}\right)}^{2}+{\left(Y_{m}-Y_{k}^{D2D}\right)}^{2}}.$$
where $X_{k}^{D2D}$ and $Y_{k}^{D2D}$ are the abscissa and ordinate of the *k*th neighbor helper, respectively. The transmission delay of the subtasks in the *m*th user to the *k*th neighbor helper is expressed as:(13)$$T_{m,k}^{t,D2D}=\frac{\sum_{n=1}^{N}\left(X_{m,n,k}D_{m,n}\right)}{R_{m,k}^{D2D}}.$$

The execution delay of the subtasks offloaded to the *k*th neighbor helper in the *m*th user is expressed as:(14)$$T_{m,k}^{e,D2D}=\frac{\sum_{n=1}^{N}\left(X_{\left(m,n,k\right)}\times D_{m,n}^{D2D}\times c\right)}{f_{k}}.$$
where $f_{k}$ is the CPU speed of the *k*th neighbor helper. The delay of offloading subtasks in the *m*th user to the *k*th neighbor helper is expressed as:(15)$$T_{m}^{D2D}=\max\left\{\left(T_{m,k}^{t,D2D}+T_{m,k}^{e,D2D}\right),k\in \left\{1,2,\ldots ,K\right\}\right\}$$

The total energy consumption of the subtasks offloaded to the neighbor helper in the *m*th user is expressed as:(16)$$E_{m}^{D2D}=P_{m}\times \sum_{k=1}^{K}T_{m,k}^{t,D2D}.$$

#### 2.2.4. Objective Function Formation

The total energy consumption of the *m*th user is expressed as:(17)$$E_{m}=E_{m}^{local}+E_{m}^{MEC}+E_{m}^{D2D}.$$

The total delay spent by the *m*th user is expressed as:(18)$$T_{m}=\max\left\{T_{m}^{local},T_{m}^{MEC},T_{m}^{D2D}\right\}.$$

The total energy consumption of all users in the scenario is expressed as:(19)$$E=(\sum_{m=1}^{M}E_{m}).$$

The average energy consumption per user is expressed as:(20)$$\bar{E_{aver}}=\frac{1}{M}\times E.$$

The total delay is expressed as:(21)$$T=\max\{T_{m},\;m\in \left\{1,2,\ldots ,M\right\}\}.$$

In this paper, our goal is to minimize the average cost of the user, where the cost refers to the weighted sum of the completion delay and average energy consumption of the user to complete all subtasks. Therefore, our objective function is expressed as follows:(22)$$\begin{array}{ccc} \underset{X,P}{min}\; & \left(\gamma_{1}T+\gamma_{2}\bar{E_{aver}}\right) & \\ s.t.\; & C1:\gamma_{1}+\gamma_{2}=1, & \\ & C2:X_{m,n,k}=\left\{0,1\right\}; & \\ & C3:\sum_{k=0}^{K+1}x_{m,n,k}=1; & \\ & C4:P_{m}\leq P_{m}^{MAX}; & \\ & C5:T_{m}\leq T_{m}^{MAX}; & \\ & C6:E_{m}\leq E_{m}^{MAX}. & \end{array}$$

In C1, $\gamma_{1}$ and $\gamma_{2}$ are the weighting factors of completion delay and average energy consumption, respectively, and their sum is 1. C2 indicates that the offloading decision value of the subtask is binary, either 0 or 1. C3 means that each subtask can only be calculated on one device and cannot be divided. C4 indicates that the transmit power of the user cannot exceed the maximum value that the user can provide. C5 and C6, respectively, indicate that the completion delay and energy consumption of each user to execute subtasks cannot exceed the maximum value that the user can tolerate. It can be known from constraint C2 that $X_{m,n,k}$ is a binary integer variable. In addition, according to Formulas (Equation 5), (Equation 9), (Equation 11), (Equation 15), (Equation 18), and (Equation 21), it can be obtained that the objective function (Equation 22) and constraints C5 and C6 are nonlinear with respect to the variables $X_{m,n,k}$ and $P_{m}$. Therefore, the objective function is a mixed integer nonlinear function.

## 3. Joint Optimization of Power Allocation Strategy and Offloading Strategy

From Section 2, we can see that the objective function of this paper is a mixed integer nonlinear problem. We first proposed an Enhanced Particle Swarm Optimization algorithm (EPSO), through the EPSO algorithm we can obtain the user’s optimal transmission power allocation strategy. Then, we utilize the Genetic Algorithm (GA) to obtain the optimal offloading strategy for the subtasks. Finally, we obtain the EPSO-GA by combining the EPSO and GA, and then obtain the optimal transmit power allocation strategy and subtask offloading strategy through the EPSO-GA.

### 3.1. Optimizing Power Allocation Strategy

In this section, we investigate how to optimize the user’s transmit power allocation strategy given the offloading strategy of the subtasks.

We propose an algorithm called EPSO to optimize the user’s transmit power. The EPSO algorithm is obtained by improving the Particle Swarm Optimization algorithm (PSO). The EPSO mainly includes the following steps: First, randomly initialize the position and velocity of the particles in the area, the individual optimal strategy, and the global optimal strategy. Then, the fitness of each particle is calculated and compared with the individual optimal strategy and the population optimal strategy, and, then, the new velocity and position are obtained according to the iterative formula until the end condition is satisfied.

In this paper, we set the population size as $population_{size}$, the position of each particle in the population represents a power allocation strategy, and the position of each particle is determined by *M* dimensions. Each dimension represents the transmit power of a user. Then, the position of the *i*th particle in the iterth generation is expressed as follows:(23)$$Position_{\left(iter,i\right)}=\left(po_{\left(iter,i,1\right)},po_{\left(iter,i,2\right)},\ldots ,po_{\left(iter,i,M\right)}\right)$$
where iter is the current iteration number. In EPSO, the position of the particle swarm is updated by providing the particle velocity. The velocity of the *i*th particle in the iterth generation is expressed as follows:(24)$$V_{\left(iter,i\right)}=\left(v_{\left(iter,i,1\right)},v_{\left(iter,i,2\right)},\ldots ,v_{\left(iter,i,M\right)}\right)$$

The individual optimal value of the iterth generation is expressed as follows:(25)$$fp_{iter}=\left(fp_{\left(iter,1\right)},fp_{\left(iter,2\right)},\ldots ,fp_{\left(iter,population_{size}\right)}\right)$$
where $fp_{\left(iter,population_{size}\right)}$ represents the individual optimal fitness of the $population_{size}$th particle of the iterth generation so far. The fitness function of the *i*th particle of the iterth generation is expressed as follows:(26)$$f_{\left(iter,i\right)}=\left(\gamma_{1}T_{aver}+\gamma_{2}\bar{E_{aver}}\right)$$

The global optimal position up to the iterth generation is expressed as follows:(27)$$fg_{iter}=min\left(fp_{1},fp_{2},\ldots ,fp_{iter}\right)$$

The velocity update formula of the *i*th particle in the (iter+1)th generation is expressed as follows:(28)$$v_{\left(iter+1,i,m\right)}=wv_{\left(iter,i,m\right)}+c_{1}r_{1}\left(iter+1,i\right)\left[fp_{\left(iter,i\right)}-f_{\left(iter,i\right)}\right]+c_{2}r_{2}\left(iter+1,i\right)\left[fg_{iter}-f_{\left(iter,i\right)}\right]$$

The location update formula of the *i*th particle in the (iter+1)th generation is expressed as follows:(29)$$po_{\left(iter+1,i,m\right)}=po_{\left(iter,i,m\right)}+v_{\left(iter+1,i,m\right)}$$
where $c_{1},c_{2}$ is the learning factor, also known as the acceleration constant. The value range of $r_{1}$,$r_{2}$ is (0,1). Compared with the traditional PSO, the EPSO realizes dynamic setting of $c_{1}$,$c_{2}$ and *w*. The value of the inertia weight *w* has a great influence on the particle swarm search for the global optimal solution, and the fixed inertia weight used in the standard PSO can easily cause the algorithm to fall into a local optimum. Therefore, we assign a larger value to *w* when the algorithm starts to execute, so that the algorithm can extensively search in the global scope of the definition domain with a larger speed step at the beginning, so that the algorithm can run faster. A large probability converges to the global optimal solution. Then, with the iteration of the algorithm, *w* is gradually reduced, so that the solution of the problem can be searched more fully near the global optimal area. In this paper, the calculation method of the weight in the algorithm is as follows.

When the current fitness is greater than the average fitness:(30)$$w\left(iter+1\right)=w_{min}-\frac{\left(w_{max}-w_{min}\right)\times \left(f_{\left(iter,i\right)}-fgiter\right)}{f_{\left(iter,aver\right)}-fgiter}$$

When the current fitness is less than the average fitness:(31)$$w\left(iter+1\right)=w_{max}+\frac{(w_{max}-w_{min})\times iter}{Iter}$$
where Iter is the maximum iteration number, $f_{\left(iter,i\right)}$ represents the fitness of the *i*th particle of the iterth generation, and $f_{\left(iter,aver\right)}$ represents the average fitness of the particles in the iter generation.

The learning factors $c_{1}$ and $c_{2}$ control the moving speed of the particles toward the individual optimal solution and the global optimal solution. Reasonable control of these two values is very important for accurately and efficiently finding the optimal solution. Different from the fixed learning factor used in the standard particle swarm optimization algorithm, this paper adopts a dynamic adjustment strategy for the learning factor. In the initial stage of the iteration, the value of $c_{1}$ is larger, and the value of $c_{2}$ is smaller, which can cause the algorithm to fully search for solutions around the particles as the algorithm iterates, gradually reduce the value of $c_{1}$, and increase the value of $c_{2}$, so that the particles approach the global optimal solution. The calculation formulas of learning factors $c_{1}$ and $c_{2}$ are as follows:(32)$$c_{1}=c_{11}-sin\left(iter\times \pi /\left(2\times Iter\right)\right).$$
(33)$$c_{2}=c_{21}+sin\left(iter\times \pi /\left(2\times Iter\right)\right).$$
where $C_{11}=2$, $C_{21}=1$.

The pseudocode of the EPSO algorithm is shown in Algorithm 1.
**Algorithm 1** EPSO**Input:***M*, *N*, *K*, $Iter_{1}$, *X*, $f_{m}$, $f_{server}$, $f_{h}$, $p_{c}$, $p_{m}$;**Output:***P*, fit; 1:Initialize the position $Position_{\left(0,i\right)}$ and velocity $V_{\left(0,i\right)}$ of each particle in the population;
 2:Initialize the individual optimal position Position_pbest(0,i) and individual optimal fitness fp(0,i) of each particle and the global optimal position Position_gbest(0) and global optimal fitness fg(0) of the population; 3:**for** iter=1 to $Iter_{1}$ **do** 4:   Calculate $f_{\left(iter,i\right)}$ with Formula (Equation 26); 5:   Record $fp_{\left(iter,i\right)}$ and Position_pbest(iter,i); 6:   Record $fg_{\left(iter\right)}$ and Position_gbest(iter); 7:   Update $V_{\left(iter,i,m\right)}$ and po(iter+1,i,m) according to Formulas (Equation 28) and (Equation 29), respectively; 8:**end for** 9:fit = $fg_{\left(iter\right)}$; 10:**return** *P*, *fit*.

Finally, we analyze the computational complexity of the EPSO algorithm. The computational complexity of the EPSO algorithm is mainly composed of the initialization phase and the loop phase. The computational complexity of the initialization phase is: $3\times O(population_{size}\times M)$. The total complexity of the loop phase is: $3\times Iter\times O(population_{size}\times M)$. Therefore, the computational complexity of the EPSO is: $3\times O(population_{size}\times M+3\times Iter\times O\left(population_{size}\times M\right)\approx \left(3\times \left(Iter+1\right)\times population_{size}\times M\right)$.

### 3.2. Optimization Subtasks Offloading Strategy

From the previous section, we can obtain the transmit power allocation strategy of users with intensive computing tasks. In this section, we use the Genetic Algorithm (GA) to optimize the offloading strategy of the subtasks. The GA is a method to search for the optimal solution by simulating the natural evolution process. The GA includes operations such as selection, crossover, mutation, etc. The crossover and mutation operations cause the genetic algorithm to have good robustness. Through the continuous evolution of the population, the population gradually converges to the direction of the optimal solution. The key terms of the GA are described below.

#### 3.2.1. Population, Chromosome, Gene

In this paper, the population represents a collection of some possible offloading strategies of subtasks. Each individual in the population is called a chromosome, each chromosome represents an offloading strategy. Chromosomes are constituted of many genes. Each gene represents an offload decision of a subtask. In this paper, our scenario has an edge server, M users, each user contains N subtasks, and each user has H-neighboring devices. Therefore, we denote a chromosome in the population as: $X_{iter,j}=\left\{a_{1},a_{2},\ldots ,a_{M*N}\right\}$, where $X_{iter,j}$ represents the *j*th chromosome in the population of the iterth generation, and $a_{1}$ is a gene on the chromosome representing the offloading decision of the first subtask. Therefore, the value range of the gene is: [0,K+1]. Among them, $a_{1}=0$ indicates that the subtask is executed locally, $a_{1}=K+1$ indicates that the subtask is offloaded to the edge server for execution, while other values indicate that the subtask is offloaded to the corresponding neighborhood Helper device for execution.

#### 3.2.2. Select

In this paper, we use the fitness function (Equation 26) to calculate the fitness value of the chromosomes in the population of each generation, and then use the roulette method to select the parent chromosomes of the next generation. The principle of the roulette wheel method is that, the better the fitness value, the greater the probability of being selected. The roulette formula is expressed as follows:(34)$$value=\frac{f_{\left(iter,i\right)}}{\sum_{i=1}^{population_{size}}f_{\left(iter,i\right)}}$$

Since the objective function is the weighted sum of average completion delay and average energy consumption in this paper, the smaller the value of the objective function, the better. Therefore, the smaller the value obtained by Formula (Equation 34), the greater the probability of it being selected as the parent chromosome.

#### 3.2.3. Crossover

Two chromosomes are randomly selected from the parental chromosomes. The number of chromosomal genes in this paper is M×N, and, then, a locus in [1,M×N] is randomly selected as the intersection of parental chromosomes. The crossover operations are performed on the parental chromosomes to exchange part of their genes with each other to form two new daughter chromosomes.

#### 3.2.4. Mutation

In order to prevent the population from maturing prematurely, it is necessary to perform mutation operations on the population. The chromosomes in the population except the elite individuals are regarded as a whole, and the genes in the population are mutated with a certain probability to generate a new allele value.

### 3.3. Alternate Optimization

In this section, we propose an algorithm based on the alternate optimization of the EPSO and GA (EPSO-GA) to realize the optimization of the transmission power allocation strategy and subtask offloading strategy. The EPSO algorithm is used to optimize the transmission power allocation strategy under the given subtask offloading. Then, based on the obtained optimal transmission power allocation strategy, the GA is used to optimize the offloading strategy of the subtasks. During the process of alternate optimization, the value of the objective function will gradually decrease until the termination condition is met, which guarantees the effectiveness of the algorithm.

The pseudocode of the (EPSO-GA) is shown in Algorithm 2; the details are described here. In Line 1, we first initialize the offloading strategy of the subtasks as a vector *X* with one row and M×N columns, and the value range of each element is [0,K+1]. The optimal power allocation strategy *P* and the best fitness $fit_{1}$ under the offloading strategy *X* are obtained through Algorithm 1 in Line 2. The optimal subtask offloading strategy *X* and optimal fitness $fit_{2}$ under the power allocation strategy *P* are obtained using the GA in Line 3. Calculate the difference of the fitness function to judge whether the end condition is satisfied in Line 4. If the end condition is not met, the operations in Lines 5–10 are performed. Finally, the optimal power allocation strategy *P*, subtask offloading strategy *X*, and the best fitness fit are returned in Line 12.
**Algorithm 2** EPSO-GA**Input:** *M*, *N*, *K*, $f_{m}$, $f_{server}$, $f_{h}$, *c*, $c_{11}$, $c_{21}$, $r_{1}$, $r_{2}$, $Iter_{2}$, $diff_{0}$, $p_{c}$, $p_{m}$;
**Output:***P*, *X*, fit;
  1:Initialize the offloading strategy of subtasks; 2:Update the user’s *P*, $fit_{1}$, with Algorithm 1; 3:Update the subtasks *X*, $fit_{2}$, with GA; 4:$diff=fit_{1}-fit_{2}$; 5:**while** $Iter_{2}>0\;and\;diff>diff_{0}$ **do** 6:   Update the user’s *P*, $fit_{1}$, with Algorithm 1; 7:   Update the user’s *X*, $fit_{2}$, with GA; 8:   $diff=fit_{1}-fit_{2}$; 9:   $Iter_{2}--$; 10:**end while** 11:$fit=fit_{2}$; 12:**return** *P*; *X*, fit;


In this section, we will analyze the computational complexity of the EPSO-GA. In Algorithm 2, the computational complexity is mainly composed of the computational complexity of the initialization, EPSO algorithm, and GA. The computational complexity of initialization is O(M×N). The computational complexity of the EPSO algorithm is $O(3\times \left(Iter+1\right)\times population_{size}\times M)$. The computational complexity of the GA is $O(population_{size}\times M\times N+\left(population_{size}-1\right)\times M\times N\times p_{c})$. Therefore, the total computational complexity of the EPSO-GA is: $O(\left(Iter+1\right)\times \left(3\times \left(Iter+1\right)\times population_{size}\times M+population_{size}\times M\times N+\left(population_{size}-1\right)\times M\times N\times p_{c}\right))$, where $p_{c}$ is the crossover rate.

## 4. Performance Evaluation

In this section, we first evaluate the performance of the EPSO algorithm compared to other baseline algorithms. We then evaluate the performance of the EPSO-GA in comparison to other baseline algorithms. Finally, we evaluate the impact of the delay weight coefficient on the EPSO-GA. In this paper, we have three performance indicators: the average completion delay of users, the average energy consumption of users, and the average cost of users. The average cost consists of the average completion delay and the average energy consumption.

### 4.1. Parameter Settings

We set the area size of the simulated environment as a circular area with a radius of 1000 meters, the edge server is located at the center of the circle. There are multiple users and multiple neighbor helpers in the scene, and they are randomly and evenly distributed in the scene. At the same time, each user contains multiple independent subtasks. The some parameters are set as follows: *M* = 60, $P_{m}^{MAX}$ = 0.5 W, $f_{m}$ = 500 MHz, $f_{h}$ = 500 MHZ, $f_{server}$ = 1.5 GHz, *B* = 10 MHz, and σ = 10${}^{-13}$.

### 4.2. Experimental Comparison Scheme

To reveal the effectiveness of the proposed EPSO algorithm and the EPSO-GA in this paper, we compare it with the following algorithms.

Particle Swarm Optimization Algorithm (PSO): The PSO can obtain the user’s transmission power allocation strategy through a particle swarm search.

Genetic Algorithm (GA): The GA can obtain the offloading strategy of subtasks through selection, crossover, and mutations operation.

Local Execution (LE): All subtasks will be executed on the device that spawned it; this execution scheme may cause excessive latency and energy consumption.

Edge Computing (EC): All subtasks are offloaded to the edge server for execution, this offloading scheme may cause channel congestion, resulting in excessive communication delays.

### 4.3. Performance Evaluation of EPSO Algorithm

Figure 2 depicts the effect of the data size of subtask, the number of subtasks, and the number of neighbor helpers on the average cost of the user. Note that there is no case where the values of delay, energy consumption, and average cost are 0 in this paper. Therefore, when it is found that the polyline coincides with the abscissa in this paper, its actual value is not 0, which is due to the visual coincidence caused by placing the polyline in the frame in proportion. As shown in Figure 2a–c.

Figure 2a depicts the effect of the subtask data size on the average cost in the case of N=10, K=8. We can see from the figure that as the size of the subtask data increases, the average cost of each user also increases. This is because, when other parameters in the scene remain unchanged, the increase in the data size of each subtask means that the data size contained in each user is increasing, so the cost required to complete the subtasks will increase. From a vertical perspective, the transmission power allocation strategy optimized by the EPSO algorithm has the smallest average cost, which is significantly better than the transmission power allocation strategy obtained using the PSO algorithm. This is because compared with the PSO algorithm, the EPSO algorithm dynamically determines the relevant parameters of the speed formula based on the fitness of particles in each generation.

Figure 2b depicts the effect of the number of subtasks in each user on the average cost. In order to ensure a single variable, we fixed the data size of each subtask at 60 Mbit, and the number of neighbor helpers at K=8. It can be seen from the figure that, as the number of subtasks in each user increases, the average cost of the user shows an increasing trend. This is because, when other parameters remain unchanged, the increase in the number of subtasks means that the amount of data in each user is increasing. Then the average completion delay and energy consumption of each user will increase, so the average cost will increase.

Figure 2c depicts the effect of the number of neighbor helpers on the average cost. In order to ensure the principle of the single variable, we set the number of subtasks in each user as N=10, and the data size of each subtask as 60 Mbit. It can be seen from the figure that changing the number of neighbor helpers has no effect on the EPSO and PSO, this is because the resources of neighbor helpers in the current environment have already met the demand, and the resources that continue to increase will still be regarded as idle resources.

### 4.4. Performance Evaluation of EPSO-GA Algorithm

#### 4.4.1. The Effect of Data Size

Figure 3 depicts the effect of the subtask data size on average completion delay, average energy consumption, and average cost in the case of N=8 and K=8, respectively. As can be seen from Figure 3, for all offloading strategies, as the data size increases, the average completion delay and the average energy consumption will increase. This is because, when the data size of the subtasks increases, the data size of each user needs to execute will increase. The average cost is the weighted sum of the average completion delay and average energy consumption, so the average cost also increases with the data size of subtask.

In Figure 3a, from a vertical perspective, the average completion delay of LE is the longest, which is caused by the limited computing power and computing resources of the local users. As the data size continues to increase, more and more subtasks will be selected for offload execution. The completion delay of EC is greater than that of EPSO. This is because the communication bandwidth in the network is limited. When all the subtasks in the scenario are offloaded to the edge server for execution, it may cause channel congestion, resulting in excessive communication delay. In addition, the processing capacity of the edge server is also limited. When there are too many tasks offloaded to the edge server, the waiting time of some tasks will be too long. The average completion delay of GA and EPSO-GA is the smallest, because EPSO-GA not only optimizes the transmit power of users but also optimizes the offloading strategy of subtasks.

In Figure 3b, from a vertical perspective, the average energy consumption of EC is the largest, because EC offloads all the tasks to the edge server for execution, and there is no efficient matching of tasks and resources in the scene. The neighbor helpers are closer to the user, and the transmission energy required to offload tasks from the local to the neighbor helper is lower. However, the GA can achieve lower transmission energy consumption by offloading some tasks to neighbor helpers that are closer. The EPSO-GA consumes the least amount of energy; this is because the EPSO-GA realizes the joint optimization of the resource allocation strategy and subtask offloading strategy, while the EPSO and GA only optimize the resource allocation strategy and subtask offloading strategy, respectively. It is worth mentioning that LE consumes the most energy and is much larger than the other offloading strategies. In order to highlight the difference between the other offloading strategies, it is not depicted in Figure 3b, and it is also treated in the following figure comparing energy consumption.

In Figure 3c, from a vertical perspective, the average cost of LE is the largest, which is due to the large computing delay and energy consumption of local computing. The average cost of the EPSO-GA is the smallest, because the EPSO-GA jointly optimizes the user’s power allocation strategy and subtask offloading strategy.

#### 4.4.2. The Effect of the Number of Subtasks

Figure 4 depicts the effect of the number of subtasks on the average completion delay, average energy consumption, and average cost when K=8 and the subtasks data size are 50 Mbit. As can be seen from Figure 4, as the number of subtasks changes from 2 to 10, the average completion delay, average energy consumption, and average cost all show an increasing trend. This is because, when the data size of subtasks remains unchanged, the increase in the number of subtasks means that the data size of each user increases.

In Figure 4a, we can see that the average completion delay of EC is greater than that of the GA and EPSO. This is because communication resources are limited and, when all subtasks are offloaded to the edge server, it may cause channel congestion and lead to long communication delay. The reasonable distributed offloading of all subtasks will be the best solution. On the one hand, the idle resources in the environment are effectively utilized to avoid load imbalance, and, on the other hand, all subtasks can be efficiently executed, as shown in the EPSO-GA.

In Figure 4b, the difference in energy consumption between the GA and EPSO-GA increases with the number of subtasks from 2 to 10. This is because, when the subtask is offloaded, the energy consumption of the subtask execution is mainly composed of the transmission energy consumption, and the transmission power consumption is affected by the transmission power. The EPSO-GA proposed in this paper not only optimizes the subtask offloading strategy but also optimizes the user’s transmit power.

In Figure 4c, as we expected, LE had the largest average cost and the EPSO-GA had the smallest average cost. In addition, with the increase in the number of subtasks, the difference in the cost of the GA and EPSO-GA to perform subtasks also increased. This proves that a reasonable transmit power allocation strategy and subtask offloading strategy are very important for the efficient execution of subtasks.

#### 4.4.3. The Effect of the Number of Resource Helpers

Figure 5 depicts the effect of the number of neighbor helpers available to each user on the average completion delay, average energy consumption, and average cost when M=60, the data size is 50 Mbit, and the weighting factor is 0.5. It can be seen from Figure 5 that the average completion delay, average energy consumption, and average cost of LE and EC do not change with the number of neighbor helpers. This is because the LE directly executes the task locally and the EC offloads tasks to edge servers for execution, meaning they have no relation to the neighbor helper during execution.

In Figure 5a, the average completion delay of the GA decreases first and then stabilizes with the increase in the number of neighbor helpers. A turning point occurs when K=2; this means that the amount of resources in the scene just meets the optimal execution requirements of the subtasks. The neighbor helpers added later still play the role of idle resources in the scene and have not been effectively utilized. The average completion delay of the GA is smaller than that of the EPSO, which shows that the subtask offloading strategy has a greater impact on the delay than the power allocation strategy. In Figure 5b, we can see that the energy consumption of the EPSO is less than that of the GA, and the curve trend of the EPSO-GA is consistent with that of the EPSO, which is because the energy consumption is mainly affected by the transmission power. In Figure 5c, we can see that the average cost of the EPSO-GA and the GA is relatively similar, because the GA has a greater impact on the EPSO-GA.

#### 4.4.4. The Effect of the Delay Weight Coefficient on the Average Cost

Figure 6 depicts the effect of changing the delay weight factor from 0.2 to 0.8 on the average cost. From a horizontal perspective, as the weight coefficient gradually increases, the average cost also increases. This is due to the sum of the weight coefficients of the average completion delay, the average energy consumption remaining unchanged (it is always 1), and the value of the average completion delay executing subtasks being greater than the value of the average energy consumption; thus, the greater the delay weight coefficient, the greater the average cost. From a vertical perspective, the average cost of the EPSO-GA is the smallest, which shows the superiority and effectiveness of the the EPSO-GA compared to other schemes.

Figure 7 depicts the effect of the number of iterations on the performance of the algorithm. It can be seen from the figure that, as the number of iterations increases, the average cost of the GA gradually decreases; it gradually stabilizes after 6th iterations, which shows that the optimal number of iterations of the GA is 6. For the PSO, EPSO, and EPSO-GA algorithms, the average cost has remained stable since the number of iterations is 2. This shows that when the number of iterations is 2, the PSO, EPSO, and EPSO-GA have converged, and the convergence speed is faster. In addition, the average cost of the EPSO-GA is the lowest, which shows that the EPSO-GA not only has a fast convergence speed but also has a low average cost. It is an effective algorithm for solving resource allocation strategy and offloading strategy.

## 5. Conclusions

In this paper, we study the user’s resource allocation strategy and subtask offloading strategy in a D2D-enabled MEC network. First, we propose the EPSO algorithm to optimize the user’s transmit power allocation strategy. Then, the offloading strategy of subtasks is optimized by using the GA. Finally, we propose the EPSO-GA to alternately iterative optimize the resource allocation strategy and subtask offloading strategy to obtain the global optimal transmit power allocation strategy and subtask offloading strategy, which minimizes the average cost of the users. The simulation results show that no matter how the simulation parameters change, such as the data size of the subtask, the number of subtasks in each user, the number of neighbor helpers, etc., the average completion delay, average energy consumption, and average cost of the EPSO-GA are always the smallest, which also proves the effectiveness of the EPSO-GA.

## Figures and Tables

**Figure 1 sensors-23-02565-f001:**
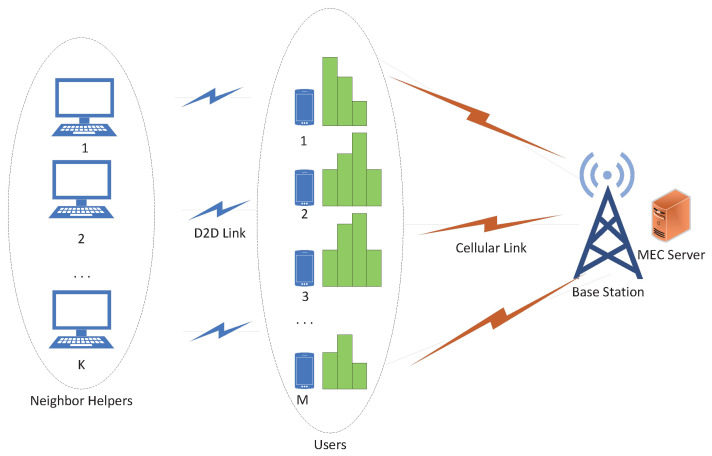
Network communication model.

**Figure 2 sensors-23-02565-f002:**
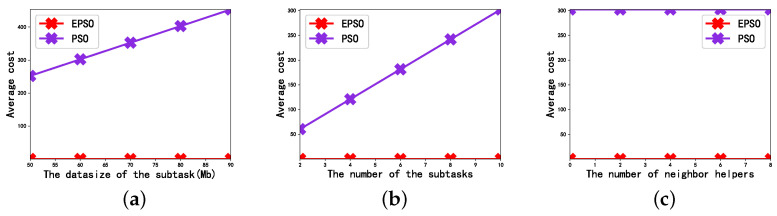
Performance evaluation of the EPSO algorithm: (**a**) The effect of data size on average cost. (**b**) The effect of number of subtasks on average cost. (**c**) The effect of number of neighbor helpers on average cost.

**Figure 3 sensors-23-02565-f003:**
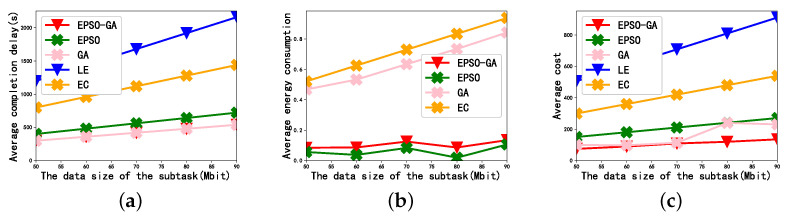
The effect of the data size of the subtask: (**a**) Average completion delay. (**b**) Average energy consumption. (**c**) Average cost.

**Figure 4 sensors-23-02565-f004:**
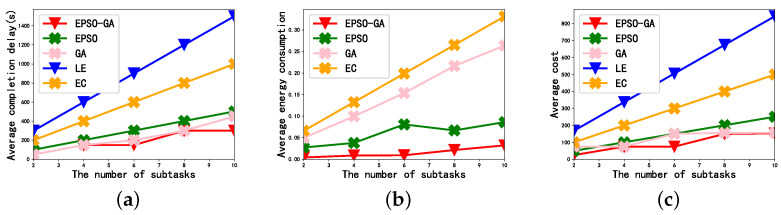
The effect of the number of subtasks: (**a**) Average completion delay. (**b**) Average energy consumption. (**c**) Average cost.

**Figure 5 sensors-23-02565-f005:**
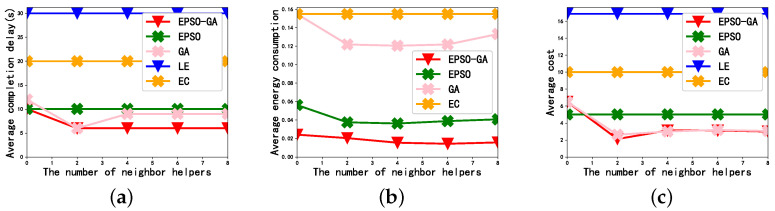
The effect of the number of resource devices: (**a**) Average completion delay. (**b**) Average energy consumption. (**c**) Average cost.

**Figure 6 sensors-23-02565-f006:**
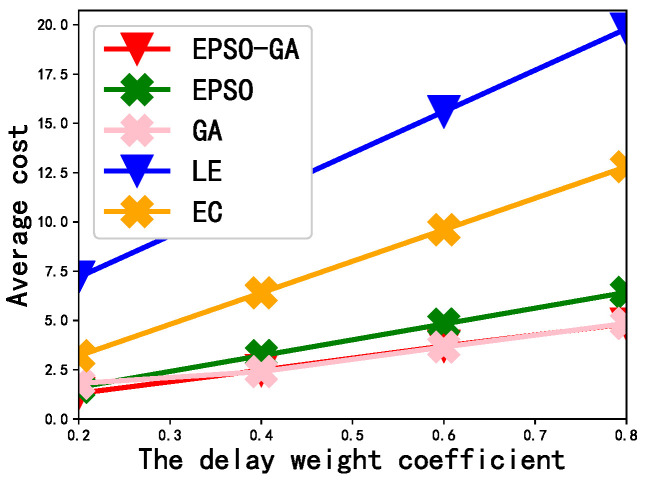
The effect of the delay weight coefficient on the average cost.

**Figure 7 sensors-23-02565-f007:**
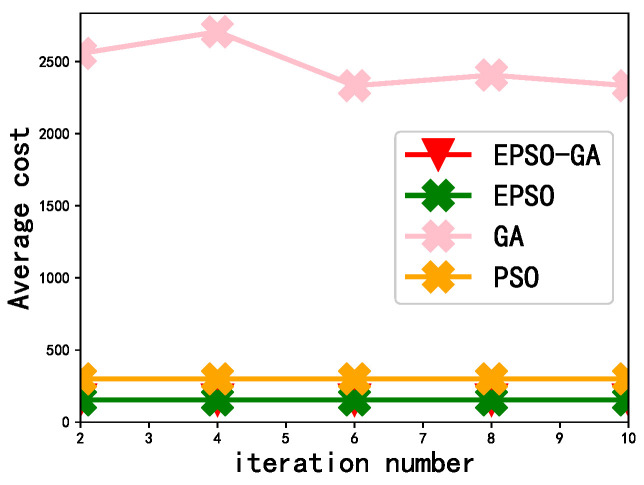
The effect of iteration number on algorithm convergence speed.

## Data Availability

Not applicable.

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
