# Peer review of "Joint Optimization of Multi-User Partial Offloading Strategy and Resource Allocation Strategy in D2D-Enabled MEC"

_sensors, 2023, doi:10.3390/s23052565_

Round 1
Reviewer 1 Report
In this paper, the authors proposed multi-user D2D offloading strategy to assist MEC networks. The optimization problem was formulated, and a solution based on PSO and GA was presented. I have the following comments regarding this paper.
1- The introduction section needs to be highly improved by highlighting the motivation and the novelty behind the presented work.
2- The optimization problem given in (22) is not complete, as the decision variables are not presented in the min equation. Also, the authors need to explain in detail why this problem is categorized as mixed non-linear integer programming.
3- The simulation figures are completely unclear and need to be highly improved.
4- There are a lot of typos and grammatical errors in the paper. To name a few:
· In p.6, the sentence “Objective Function Formation The total energy consumption” is completely vague.
· In p.9, l.270 “Ther through”
· In p .10 l. 285 “Select In this paper”
· In p.10, l.294, “”Crossover Two”
Author Response
We sincerely thank the reviewer for the valuable and insightful comments/suggestions to improve the quality of this manuscript. Our responses to each suggestion are as follows:
Point 1: The introduction section needs to be highly improved by highlighting the motivation and the novelty behind the presented work.
Response 1: Thank you very much for your comments. In the revised manuscript, we have revised the “introduction” section. Based on a thorough survey of the related works, we have found that there are few studies on offloading strategy and resource allocation strategy for multiple users in the D2D-MEC communication model, especially when each user contains multiple subtasks. And in real life, it is very common for a user to contain subtasks. Therefore, we believe that it is necessary to study the subtasks offloading strategy and users power allocation strategy of users with subtasks under the D2D-MEC communication model. On page 2, line 71 to page 2, Line 90 presents the innovation of this paper.
Point 2: The optimization problem given in (22) is not complete, as the decision variables are not presented in the min equation. Also, the authors need to explain in detail why this problem is categorized as mixed non-linear integer programming.
Response 2:
In the revised paper, we modified the objective function in Equation 22. Particularly, we have changed T_aver to T, and added optimization variables X and P, as shown in Equation 22.
In addition, it can be known from constraint C2 that X_{m,n,k} is a binary integer variable. And according to formulas (5), (9), (11), (15), (18), (21), it can be obtained that the objective function (22) and constraints C5, C6 are nonlinear with respect to the variables X_{m,n,k} and P_m. we also analyze the reason why the objective function is a mixed integer nonlinear function in lines 183 to 187 on page 6.
Point 3: The simulation figures are completely unclear and need to be highly improved.
Response 3: In the revised paper, we have increased the font size and line width of all figures in the paper.
Point 4: There are a lot of typos and grammatical errors in the paper. To name a few:
- In p.6, the sentence “Objective Function Formation The total energy consumption” is completely vague.
- In p.9, l.270 “Ther through”
- In p .10 l. 285 “Select In this paper”
- In p.10, l.294, “Crossover Two”
Response 4: Thanks for your comments. We have corrected the mentioned typo errors in the paper. The paper is also corrected by a native speaker.
We have uploaded our modified paper as an attachment. For convenience, all changes are marked in red in the modified paper.
Please refer to the attachment.

Reviewer 2 Report
- The equation number 22: the time and the energy have different scales even if the average is considered. They should be re-scaled to [0,1] interval by dividing by the maximal value for example
- in the equation 21 the total de;ay is expressed and in equation 22 the average is considered instead.
- the problem with considering subtasks is the schedule. The completion constraints between successive tasks should be taken into consideration
- The number of users is very small (equal to 3). Large number of users should be considered. Same remark for 3b-3c
- The converge speed of the metaheuristics should be discussed.
- all the references are IEEE
- the setting parameters foir the metaheuristics are missing. How these algorithms were tuned?
- Figure 2: the cost is null? if not a zoom should be added to the figure
Author Response
We sincerely thank the reviewer for the valuable and insightful comments/suggestions to improve the quality of this manuscript. Our responses to each suggestion are as follows:
Point 1: The equation number 22: the time and the energy have different scales even if the average is considered. They should be re-scaled to [0,1] interval by dividing by the maximal value for example.
Response 1: Thank you very much for your constructive suggestion. In the existing work, most of the papers that considers the delay and energy consumption as the objective function, and the delay and energy consumption are formulated as weighted sum, such as [1], [2], [3]. In [1], the objective function is the weighted sum of task completion delay and energy consumption. Latency and energy consumption are not scaled. In [2], the objective functions is a weighted sum of delay and energy consumption. Latency and energy consumption are not scaled. In [3], the sum of weighted coefficients is 1 and the objective functions are the weighted sum of delay and energy consumption, and there is no time delay scale with energy consumption. The objective function of this article is also made by referring to the above paper. But we think that "scaling the delay and energy consumption into the same scale for calculation" may be a more effective measurement method, and we will consider applying it in future work.
Point 2: In the equation 21 the total delay is expressed and in equation 22 the average is considered instead.
Response 2: Thank you very much for your suggestion. In the previous manuscript, due to negligence, the total delay T in formula (22) was written as the average delay T_aver, and now T_aver in formula (22) has been changed to T. Formula (21) is correct.
Point 3: The problem with considering subtasks is the schedule. The completion constraints between successive tasks should be taken into consideration
Response 3:
Thank you very much for your suggestions. From the perspective of the dependencies of the subtasks, we set all subtasks to be independent of each other in line 109 on page3 of this paper, so there is no completion constraint between subtasks. We will investigate offloading strategy for subtasks with dependencies in future work.
From the perspective of device computing capabilities, we assume that each device can only execute one subtask at a time in line 126 to line 128 on page 4 of this paper. If there are multiple subtasks that are offloaded to the same device at the same time, they will be queued for execution, the next subtask can only be executed when the previous subtask is completed.
Point 4: The number of users is very small (equal to 3). Large number of users should be considered. Same remark for 3b-3c
Response 4: Thank you very much for your suggestion. The number of users in the algorithm performance evaluation in this paper is changed to 60, as shown in line 312 on page 11, and the performance of each algorithm is re-tested when the number of users is 60. (Remarks: Because this paper considers the case that users contain subtasks, the number of variables in the offloading decision is the product of the number of users and the number of subtasks: M*N.)
The figure 3(c) has been modified.
Point 5: The converge speed of the metaheuristics should be discussed.
Response 5: Thank you very much for your suggestion. We have added a discussion of the convergence rate of the algorithm in the newly submitted manuscript, as shown in lines 448 to 457 on page 14 and Figure 7.
Point 6: all the references are IEEE.
Response 6: Thank you very much for your suggestion. In the latest submitted manuscript, we have added 4 new papers, three of which are from MDPI's sensors journal, and one from Elsevier's Future Generation Computer Systems journal, as shown in [11], [12], [19], [20] in the references section.
Point 7: the setting parameters for the metaheuristics are missing. How these algorithms were tuned?
Response 7: Thank you very much for your suggestion. In the algorithm EPSO, we changed "while iter <= Iter" to "for iter=1 to Iter", which solved the problem that iter has no initial value.
The execution of the algorithm in this paper is shown in the EPSO-GA algorithm, which will be described as below:
Step 1: Initialize the offloading strategy X of the subtasks.
Step 2: Bring X into the EPSO algorithm, and use the EPSO algorithm to optimize the user's transmission power allocation strategy to obtain P and fit1.
Step 3: Bring P into the GA, and use the GA to optimize the offloading strategy of the subtasks to obtain X and fit2.
Step 4: Make a difference between fit1 and fit2 and judge whether the difference meets the termination condition. If not, repeat steps 2, 3, and 4 until the termination condition is met. Finally, the best subtasks offloading strategy X, the best users power allocation strategy P, and the best fitness fit are obtained.
Point 8: Figure 2: the cost is null? if not a zoom should be added to the figure.
Response 8: Thank you very much for your suggestion. We have made modifications to Figures 3, 4, 5, and 6 in this paper. In addition, in Figure 2, due to the large difference between the maximum value and the minimum value, even if the EPSO line is zoomed in, the EPSO line almost coincides with the abscissa axis visually. Therefore, for Figure 2, we made the following remarks in lines 327 to 331 on page 11: ” Note that there is no case where the values of delay, energy consumption and average cost are 0 in this paper. Therefore, when it is found that the polyline coincides with the abscissa in this paper, its actual value is not 0, which is due to the visual coincidence caused by placing the polyline in the frame in proportion. As shown in Figure 2. "
[1] Fang T, Yuan F, Ao L, et al. Joint task offloading, D2D pairing, and resource allocation in device-enhanced MEC: a potential game approach[J]. IEEE Internet of Things Journal, 2021, 9(5): 3226-3237.
[2] Wang H, Lin Z, Lv T. Energy and delay minimization of partial computing offloading for D2D-assisted MEC systems[C]//2021 IEEE Wireless Communications and Networking Conference (WCNC). IEEE, 2021: 1-6.
[3] Yan J, Bi S, Zhang Y J, et al. Optimal task offloading and resource allocation in mobile-edge computing with inter-user task dependency[J]. IEEE Transactions on Wireless Communications, 2019, 19(1): 235-250.
We have uploaded our modified paper as an attachment. For convenience, all changes are marked in red in the modified paper.
Please refer to the attachment.

Round 2
Reviewer 1 Report
The authors fulfilled all the required modifications
Reviewer 2 Report
all the comments were addressed.